# Sonic hedgehog-dependent recruitment of GABAergic interneurons into the developing visual thalamus

Rachana Deven Somaiya[1,2], Katelyn Stebbins[1,2,3], Ellen C Gingrich[4,5], Hehuang Xie[6,7,8,9], John N Campbell[10,11], A Denise R Garcia[4,5], Michael A Fox[1,7,12,13]*

[1]Center for Neurobiology Research, Fralin Biomedical Research Institute at Virginia Tech Carilion, Roanoke, United States; [2]Graduate Program in Translational Biology, Medicine, and Health, Virginia Tech, Blacksburg, United States; [3]Virginia Tech Carilion School of Medicine, Roanoke, United States; [4]Department of Biology, Drexel University, Philadelphia, United States; [5]Department of Neurobiology and Anatomy, Drexel University College of Medicine, Philadelphia, United States; [6]Fralin Life Sciences Institute at Virginia Tech, Blacksburg, United States; [7]School of Neuroscience, College of Science, Virginia Tech, Blacksburg, United States; [8]Genetics, Bioinformatics and Computational Biology Program, Virginia Tech, Blacksburg, United States; [9]Department of Biomedical Sciences and Pathobiology, Virginia–Maryland College of Veterinary Medicine, Blacksburg, United States; [10]Department of Biology, University of Virginia, Charlottesville, United States; [11]Neuroscience Graduate Program, University of Virginia, Charlottesville, United States; [12]Department of Biological Sciences, College of Science, Virginia Tech, Blacksburg, United States; [13]Department of Pediatrics, Virginia Tech Carilion School of Medicine, Roanoke, United States

*For correspondence: mafox1@vt.edu

## Abstract

Axons of retinal ganglion cells (RGCs) play critical roles in the development of inhibitory circuits in visual thalamus. We previously reported that RGC axons signal astrocytes to induce the expression of fibroblast growth factor 15 (FGF15), a motogen required for GABAergic interneuron migration into visual thalamus. However, how retinal axons induce thalamic astrocytes to generate *Fgf15* and influence interneuron migration remains unknown. Here, we demonstrate that impairing RGC activity had little impact on interneuron recruitment into mouse visual thalamus. Instead, our data show that retinal-derived sonic hedgehog (SHH) is essential for interneuron recruitment. Specifically, we show that thalamus-projecting RGCs express SHH and thalamic astrocytes generate downstream components of SHH signaling. Deletion of RGC-derived SHH leads to a significant decrease in *Fgf15* expression, as well as in the percentage of interneurons recruited into visual thalamus. Overall, our findings identify a morphogen-dependent neuron–astrocyte signaling mechanism essential for the migration of thalamic interneurons.

## Editor's evaluation

This study address an interesting mechanistic question with important implications for fundamental neural development. The authors' findings support a model in which retinal ganglion cell axons secrete Shh in the visual thalamus to induce FGF15 expression by astrocytes, which in turn attracts migrating Gad1-expressing cells (interneurons) into the vLGN and dLGN during mouse development. Interestingly, neuronal activity is not required for this process. These findings will be appreciated by a wide range of developmental neuroscientists.

## Introduction

The retina receives light-derived signals from our visual environment and transmits those signals to several dozen regions of the brain via axons of retinal ganglion cells (RGCs). In rodents, one of the brain regions densely innervated by RGC axons is the visual thalamus, which includes several retino-recipient nuclei, such as the dorsal lateral geniculate nucleus (dLGN), intergeniculate leaflet (IGL), and ventral lateral geniculate nucleus (vLGN). Despite being adjacent, these nuclei have diverse roles in visual processing, with the dLGN being important for image-forming visual functions, and the vLGN and IGL contributing more to non-image-forming visual functions (such as visuomotor functions, circadian photoentrainment, and mood regulation) (*Fratzl et al., 2021*; *Guido, 2018*; *Huang et al., 2019*; *Monavarfeshani et al., 2017*; *Salay and Huberman, 2021*). Not surprisingly based on these diverse functions, the principal neurons and their connectivity differ greatly between dLGN and vLGN/IGL (*Krahe et al., 2011*; *Sabbagh et al., 2018*; *Sabbagh et al., 2021*). In fact, while the most abundant neurons in dLGN are glutamatergic thalamocortical relay cells, most neurons in vLGN are GABAergic and appear to represent a heterogeneous group of cell types with distinct morphologies, projections, and functions (*Ciftcioglu et al., 2020*; *Crombie and Busse, 2021*; *Harrington, 1997*; *Yuge et al., 2011*).

Despite these differences, there are some similarities in the cell types of dLGN and vLGN, including the presence and distribution of local interneurons (*Arcelli et al., 1997*; *Evangelio et al., 2018*; *Jaubert-Miazza et al., 2005*). In fact, a shared mechanism appears to exist that contributes to the recruitment and integration of these GABAergic neurons into both dLGN and vLGN (*Golding et al., 2014*; *Su et al., 2020*). Specifically, the recruitment of migrating interneurons into the perinatal dLGN and vLGN requires the innervation of these regions by RGC axons. Our previous work identified a mechanism underlying this process (*Su et al., 2020*). We reported that the expression of migratory cue fibroblast growth factor 15 (FGF15) by thalamic astrocytes is dependent on retinal inputs and the loss of this FGF leads to a reduction in the number of GABAergic interneurons in visual thalamus. The decreasd number of GABAergic interneurons in the absence of FGF15 is not due to an increase in programmed cell death. Instead, these cells are misrouted into regions adjacent to the dorsal thalamus, specifically affecting their recruitment into dLGN and vLGN. However, it remains unclear how exactly retinal axons instruct astrocytes to generate FGF15 and to facilitate interneuron recruitment into visual thalamus.

Here, we sought to determine the role of retinal activity and axon-derived factors in the FGF15-dependent interneuron migration into the developing visual thalamus. Our data show that impairing the activity of RGCs by genetically expressing tetanus toxin (TeNT) disrupts eye-specific segregation in dLGN and causes visual deficits in mice; however, it had little impact on interneuron recruitment into visual thalamus. Instead, we discovered that retinal-derived sonic hedgehog (SHH) is necessary for interneuron recruitment. Not only do RGCs express SHH in the perinatal retina, but also astrocytes in the developing visual thalamus express several downstream molecules involved in the canonical SHH signaling pathway, such as *Ptch1*, *Smo*, and *Gli1*. Conditional deletion of SHH from RGCs led to decrease in *Fgf15* expression and deficits in interneuron migration into visual thalamus. Our findings demonstrate a novel activity-independent and SHH-dependent molecular mechanism for RGC axons to orchestrate thalamic interneuron migration.

## Results

### SHH, but not retinal activity, is critical for interneuron recruitment into visual thalamus

Using *Atoh7⁻/⁻* (also called *Math5⁻/⁻*) mutant mice, which lack RGC inputs to visual thalamus (*Figure 1A*), we previously reported a significant reduction in the number of interneurons in visual thalamus (*Su et al., 2020*). Here, we demonstrate this loss using in situ hybridization (ISH) for *Gad1* mRNA, which encodes for glutamic acid decarboxylase 67 (GAD67), an enzyme required for the production of GABA. This confirmed significant reduction in *Gad1⁺* neurons in both dLGN and vLGN (*Figure 1A–C*). It is important to highlight that the reduction in *Gad1⁺* cells in *Atoh7⁻/⁻* vLGN is less striking than in dLGN, since most *Gad1⁺* cells in vLGN are principal projection cells and not local interneurons.

Although the expression of FGF15 in visual thalamus is dependent on retinal inputs (*Su et al., 2020*, *Figure 1—figure supplement 1A*), it remains unclear whether this is due to neuronal activity or

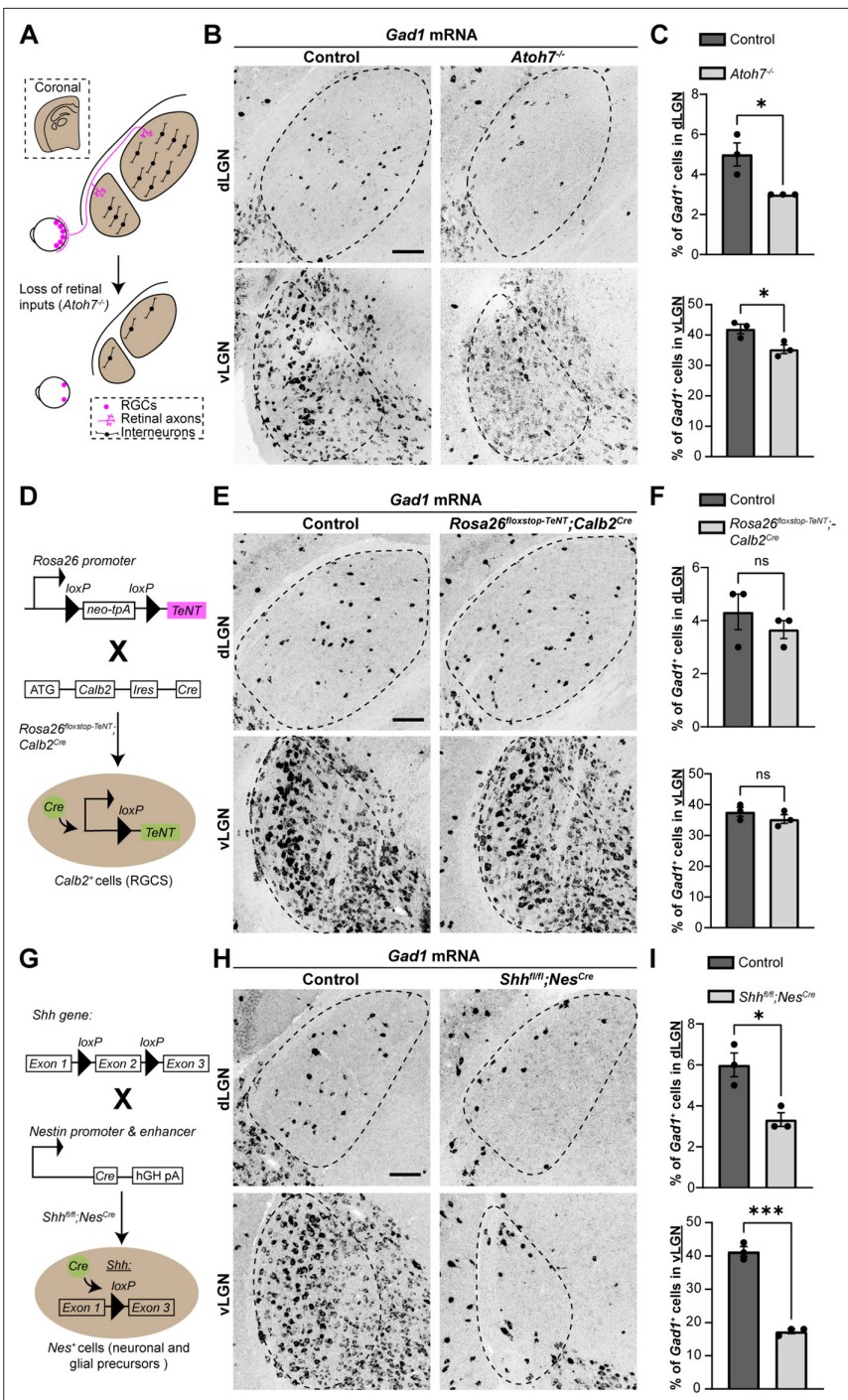

**Figure 1.** Sonic hedgehog (SHH), but not retinal activity, is required for the recruitment of interneurons into visual thalamus. (**A**) Schematic representation of loss of retinal inputs and interneurons in visual thalamus of *Atoh7*$^{-/-}$ mice. (**B**) In situ hybridization (ISH) shows a reduction in *Gad1*$^+$ cells in dorsal lateral geniculate nucleus (dLGN) and ventral lateral geniculate nucleus (vLGN) of >P150 *Atoh7*$^{-/-}$ mice compared with controls. (**C**) Quantification of percentage of *Gad1*$^+$ cells in dLGN and vLGN of >P150 controls and *Atoh7*$^{-/-}$ mice. Each data point represents one biological replicate and bars depict means ± standard error of the mean (SEM). Asterisks (*) indicate p < 0.05 by Student's *t*-test (*n* = 3 mice for each group). (**D**) Schematic representation of *Calb2*$^{Cre}$-inducible expression of tetanus toxin (TeNT) in retinal ganglion cells (RGCs). *Rosa26*$^{floxstop-TeNT}$ mice with the construct containing a *loxP*-flanked neomycin (Neo) cassette and TeNT coding sequence under *Rosa26* locus, were crossed with *Calb2*$^{Cre}$ mice that harbor a *Cre* recombinase and internal ribosome entry site (IRES) in the *Calb2* locus. (**E**) ISH for *Gad1* in dLGN and vLGN of P120 control and *Rosa26*$^{floxstop-TeNT}$;*Calb2*$^{Cre}$ mice. (**F**) Quantification revealed no significant difference

*Figure 1 continued on next page*

*Figure 1 continued*

in *Gad1+* cells in visual thalamus of control and *Rosa26floxstop-TeNT;Calb2Cre* mice. Each data point represents one biological replicate and bars depict means ± SEM. ns indicates no significant differences by Student's *t*-test (*n* = 3 mice for each group). (**G**) Schematic representation of strategy to delete SHH from neural cells in the developing brain. This was achieved by crossing *Shhfl/fl* mice, which have two *loxP* sites flanking exon 2 of the *Shh* gene, with *NesCre* transgenic mice that contain a *Cre* recombinase and human growth hormone polyadenylation signal (hGH pA) under the control of *Nestin* promoter and enhancer. (**H**) ISH revealed a dramatic reduction in *Gad1+* cells in dLGN and vLGN of P18 *Shhfl/flNesCre* mice compared with controls. (**I**) Quantification of percentage of *Gad1+* cells in dLGN and vLGN of P18 control and *Shhfl/flNesCre* mice. Each data point represents one biological replicate and bars depict means ± SEM. Asterisks represent significant differences (\*\*\*p < 0.001; \*p < 0.05) by Student's *t*-test (*n* = 3 mice for each group). Scale bars in C, E, H: 100 μm.

The online version of this article includes the following figure supplement(s) for figure 1:

**Figure supplement 1.** Thalamic *Fgf15* expression is dependent on retinal inputs and sonic hedgehog (SHH).

**Figure supplement 2.** Anatomical and functional characterization of *Rosa26floxstop-TeNT;Calb2Cre* mice.

---

developmentally regulated factors released by RGC axons. Here, we set out to distinguish between these possibilities. First, we tested whether retinal activity was necessary for the interneuron recruitment into visual thalamus. To impair neuronal activity from RGCs, we took advantage of a transgenic Cre-inducible system to express TeNT in RGCs (*Rosa26floxstop-TeNT*). TeNT is a protease that cleaves the vesicular SNARE Synaptobrevin2/VAMP2 (Syb2), which is required for the exocytosis of neurotransmitters (*Link et al., 1992*; *Schoch et al., 2001*). We crossed mice carrying the *Rosa26floxstop-TeNT* allele with *Calb2Cre* mice, in which most, if not all, RGCs express Cre recombinase (*Kerr et al., 2019*; *Sando et al., 2017*; *Zhang et al., 2008*) and widespread expression of Cre occurs in the GCL of the retina as early as P0 (*Figure 1D* and *Figure 1—figure supplement 2A*). Our selection of *Calb2Cre* over *Slc17a6Cre* (also called vesicular glutamate transporter 2, *Vglut2Cre*), another widely used transgenic line to target RGCs (*Wang et al., 2020*), was based on the relatively low expression of *Calb2* mRNA in dLGN (*Ahmadlou et al., 2018*) as compared to Slc17a6 (*Land et al., 2004*).

Studies that have employed *Rosa26floxstop-TeNT* to block neural activity have confirmed the suppression of neurotransmitter release by observing a loss of Syb2 in Cre-expressing cells (*Sando et al., 2017*). However, retinal inputs account for only 5–10% of the total synapses onto a dLGN relay cell (*Bickford et al., 2010*; *Van Horn et al., 2000*; *Sherman and Guillery, 2002*), making it difficult to detect significant changes in just a small fraction of the synapses in visual thalamus. Therefore, we sought other approaches to confirm impaired retinogeniculate (RG) neuronal activity in *Rosa26floxstop-TeNT;Calb2Cre*. RGC activity during perinatal development is critical for eye-specific segregation of retinal axons (*Huberman et al., 2003*; *Penn et al., 1998*; *Pfeiffenberger et al., 2005*), therefore, we assessed eye-specific RGC projections in *Rosa26floxstop-TeNT;Calb2Cre* mice. Retinal terminals were labeled by intraocular injection of different fluorescently conjugated Cholera Toxin Subunit B (CTB) into each eye at P12–P20. In *Rosa26floxstop-TeNT;Calb2Cre* mice, we observed a significant increase in the area occupied by inputs from both eyes, a hallmark of impaired activity-dependent RG refinement (*Figure 1— figure supplement 2D, E*). Blocking retinal activity in these mice should also significantly impact visual behaviors, therefore, we also tested responses to a 'looming stimulus' (*Koehler et al., 2019*). When presented with dark looming stimuli, control mice displayed an escape response, running to a shelter within the arena. In contrast, *Rosa26floxstop-TeNT;Calb2Cre* mutants had impaired escape responses, indicating deficits in their ability to identify visual stimuli, although it is important to note that these responses were not as impaired as observed in *Atoh7−/−* mutants (*Figure 1—figure supplement 2B, C*). Together, these sets of both results suggest that there is a significant impairment in glutamate release by retinal axons in *Rosa26floxstop-TeNT;Calb2Cre* mice, confirming that this mutant line can be used to study the role of retinal activity in the developing brain.

Using riboprobes against *Gad1*, we performed ISH to detect changes in percentage of interneurons in the visual thalamus of *Rosa26floxstop-TeNT;Calb2Cre* mice. Surprisingly, in dLGN and vLGN, we did not observe a significant difference in the percentage of *Gad1*-expressing cells between the controls and mutants, revealing that decreasing retinal activity, or at least glutamate release by RGCs, had little impact on the recruitment of interneurons into visual thalamus (*Figure 1E, F*).

Since retinal activity did not appear critical for interneuron recruitment into visual thalamus, we next investigated whether RGCs released factors beyond neurotransmitters, which could influence

migration of thalamic interneurons. SHH signaling widely regulates FGF15 expression in the embryonic brain (*Ishibashi and McMahon, 2002*; *Martinez-Ferre et al., 2016*; *Saitsu et al., 2005*) and RGCs have been reported to generate and secrete SHH (*Dakubo et al., 2008*; *Peng et al., 2018*; *Traiffort et al., 2001*; *Wallace and Raff, 1999*), suggesting that SHH might regulate FGF15 expression and interneuron migration into visual thalamus. To test whether this was the case, we utilized a conditional allele of *Shh^{fl/fl}* and crossed it with *Nes^{Cre}*, in which there is early expression of Cre in neuronal and glial progenitors (*Tronche et al., 1999*; *Figure 1G*). As previously reported, *Shh^{fl/fl}Nes^{Cre}* mutants survive gestation, however, die by P18 (*Machold et al., 2003*; *Xu et al., 2005*). In dLGN and vLGN of these mutants, we observed a significant and dramatic decrease in the percentage of *Gad1*-expressing neurons (*Figure 1H, I*). Furthermore, loss of SHH signaling in *Shh^{fl/fl}Nes^{Cre}* mice resulted in a reduction in *Fgf15* expression, similar to what was observed in mice lacking retinal inputs (*Figure 1—figure supplement 1B, C*). Thus, SHH signaling appears critical for both thalamic *Fgf15* expression and the recruitment of interneurons into dLGN and vLGN.

## Astrocytic expression of SHH signaling components

Activation of the SHH pathway is initiated when extracellular SHH binds to its canonical receptor, Patched-1 (PTCH1), present on target cells. This binding leads to the release of PTCH1-mediated inhibition of the membrane-spanning receptor, Smoothened (SMO), thereby allowing it to activate glioma-associated oncogene (GLI) transcription factors. Since our previous studies revealed that FGF15 is specifically generated by thalamic astrocytes during perinatal development (*Su et al., 2020*), we asked whether these astrocytes express components of the SHH signaling.

Using previously generated RNAseq datasets, we first assessed the expression profiles of several SHH signaling genes in the developing mouse dLGN and vLGN (*Monavarfeshani et al., 2018*). Analysis of these data showed that three canonical SHH signaling components, *Ptch1*, *Smo*, and *Gli1*, are expressed in visual thalamus at P3 (*Figure 2A*), when *Fgf15* expression is high in this region (*Su et al., 2020*). To determine whether astrocytes express these components in visual thalamus, we undertook several approaches. First, we utilized *Rosa26^{tdT};Gli1^{CreER}* mice whereby tamoxifen administration leads to Cre-mediated recombination of the tdT fluorescent protein in *Gli1*-expressing cells (*Hill et al., 2019*). Thalamic slices from P7 *Rosa26^{tdT};Gli1^{CreER}* showed the presence of *Gli1-tdT^+* cells in both dLGN and vLGN (*Figure 2B*). Immunostaining for S100ß, a commonly used marker for astrocytes, revealed S100ß expression in *Gli1-tdT^+* cells, suggesting that these astrocytes generate *Gli1* (*Figure 2C, D*). However, we recently reported that S100ß immunohistochemistry (IHC) also labels a small population of microglia in visual thalamus (*Somaiya et al., 2021*), therefore, we confirmed that *tdT^+* cells were astrocytes by ISH for *Gja1*, the gene encoding Connexin 43. In fact, we observed that *Gja1* mRNA was present in all *Gli1-tdT^+* cells, suggesting expression of this transcription factor was specific to astrocytes in visual thalamus (*Figure 2E, F*).

We next examined the expression of *Ptch1* and *Smo* in thalamic astrocytes. For this, we performed ISH in *Aldh1l1-GFP* mice in which most thalamic astrocytes are labeled with GFP (*Somaiya et al., 2021*). ISH with riboprobes against *Ptch1* and *Smo* revealed that a high percentage of *Aldh1l1-GFP^+*astrocytes express these genes in P3 dLGN and vLGN (*Figure 2G–J*). Together, these data demonstrate that astrocytes in perinatal visual thalamus have the cellular machinery to respond to SHH.

## SHH is generated by RGCs in the developing perinatal retina

Several studies have shown that RGCs generate and release SHH (*Dakubo et al., 2008*; *Peng et al., 2018*; *Su et al., 2020*; *Traiffort et al., 2001*; *Wallace and Raff, 1999*), therefore, we next asked whether they also express SHH at times corresponding to FGF15 expression in the visual thalamus. At P3, not only is *Shh* mRNA expression restricted to the GCL of the retina, but it is expressed by CALB2^+ RGCs (*Figure 3A, B*). Since at least 40 transcriptionally distinct types of RGCs exist in the mouse retina (*Tran et al., 2019*), we sought to identify whether all RGCs generate *Shh* mRNA (which looked to be the case based on ISH, *Figure 3A*) or whether specific RGC subtypes generate this morphogen in the perinatal retina. Rheaume et al. recently performed a comprehensive transcriptomic analysis of all mouse RGC subtypes at P5, therefore, we reanalyzed their publicly available single-cell RNAseq dataset to answer this question (*Rheaume et al., 2018*). Two important observations were made through this analysis: (1) all RGC subtypes express high levels of *Calb2*, confirming the reliability of

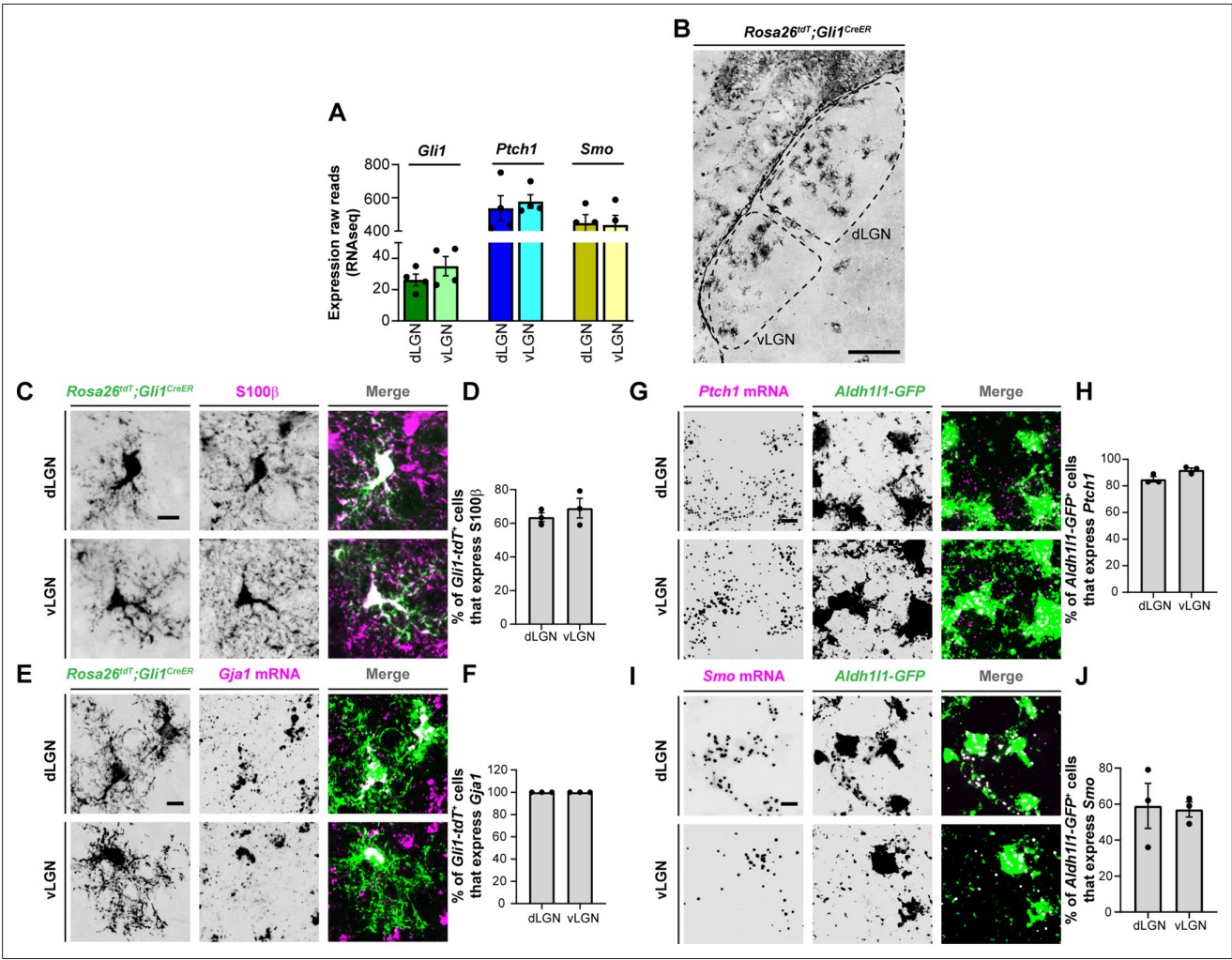

**Figure 2.** Expression of sonic hedgehog (SHH) signaling molecules by astrocytes in visual thalamus. (**A**) Raw transcript reads of mRNAs for downstream SHH signaling components in P3 dorsal lateral geniculate nucleus (dLGN) and ventral lateral geniculate nucleus (vLGN) by RNAseq. Each data point represents a different biological replicate and bars depict means ± standard error of the mean (SEM). (**B**) Presence of *Gli1-tdT*+ cells in dLGN and vLGN of P7 *Rosa26tdT;Gli1CreER* mice. (**C**) IHC for S100ß in P7 *Rosa26tdT;Gli1CreER* mice revealed S100ß expression in *Gli1-tdT*+ cells in visual thalamus. (**E**) Quantification of percentage of *Gli1-tdT*+ cells that express S100ß in P7 visual thalamus. Each data point represents one biological replicate and bars depict means ± SEM (*n* = 3 mice for each region). (**D**) In situ hybridization (ISH) for *Gja1* in P7 *Rosa26tdT;Gli1CreER* mice revealed expression of *Gja1* mRNA by *Gli1-tdT*+ cells in visual thalamus. (**F**) Quantification shows 100% of *Gli1-tdT*+ cells express the astrocytic marker *Gja1* in P7 visual thalamus. Each data point represents one biological replicate and bars depict means ± SEM (*n* = 3 mice for each region). (**G**) RNAscope-based ISH detected *Ptch1* mRNA in dLGN and vLGN of P3 *Aldh1l1-GFP* mice. This revealed expression of *Ptch1* mRNA in the cell bodies as well as in the processes of *Aldh1l1-GFP*+astrocytes. (**H**) Quantification of percentage of *Aldh1l1-GFP*+astrocytes that express *Ptch1* mRNA in P3 visual thalamus. Each data point represents one biological replicate and bars depict means ± SEM (*n* = 3 mice for each region). (**I**) RNAscope-based ISH detected *Smo* mRNA in dLGN and vLGN of P3 *Aldh1l1-GFP* mice. This revealed expression of *Smo* mRNA in the cell bodies as well as in the processes of *Aldh1l1-GFP*+ astrocytes.(**J**) Quantification of percentage of *Aldh1l1-GFP*+astrocytes that express *Smo* mRNA in P3 visual thalamus. Each data point represents one biological replicate and bars depict means ± SEM (*n* = 3 mice for each region). Scale bars in B: 200 µm and in C, E, G, I: 10 µm.

targeting this molecule to study RGCs (**Kerr et al., 2019**); (2) at least 39 out of 41 RGC subtypes express *Shh* mRNA (**Figure 3C**). These results suggested that RGCs in the developing postnatal retina express SHH, which can potentially influence the role of thalamic astrocytes.

## RGC-derived SHH is required for FGF15 expression and interneuron recruitment into visual thalamus

Previous studies have shown that RGC-derived SHH protein travels all the way to the optic nerve (**Wallace and Raff, 1999**) and can even be detected in retinorecipient brain areas (**Traiffort et al.,**

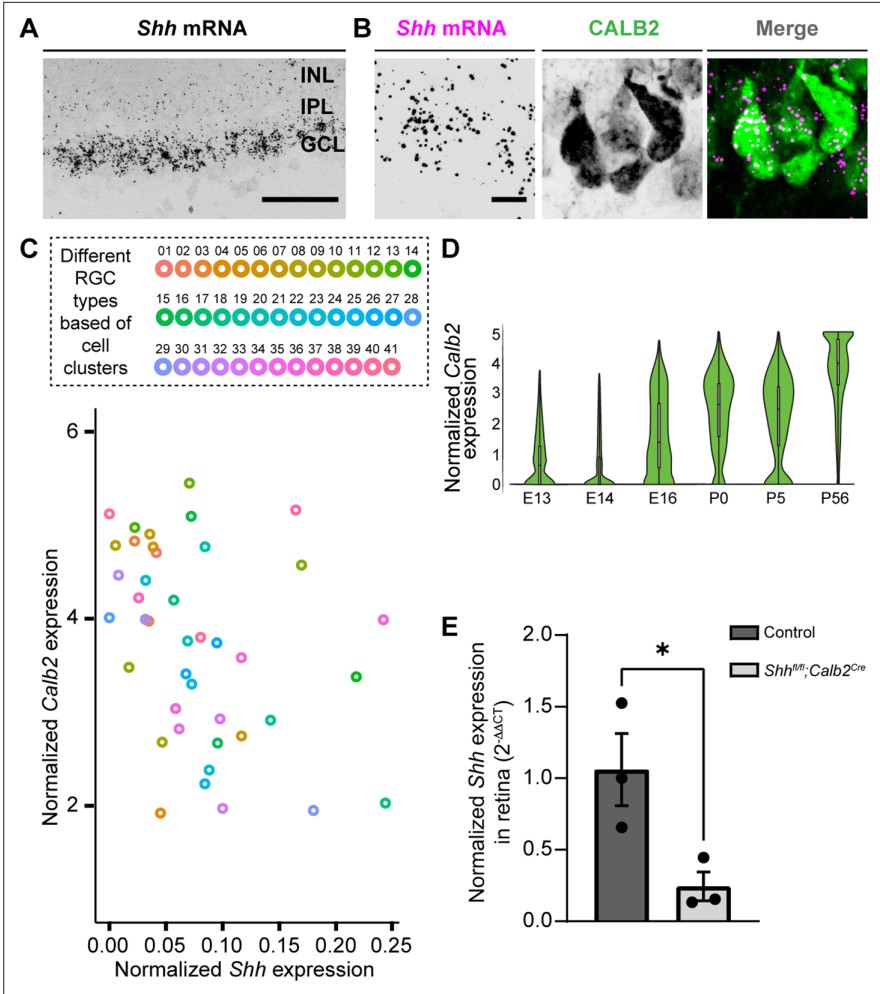

**Figure 3.** Deletion of sonic hedgehog (SHH) from retinal ganglion cells (RGCs) in the perinatal retina. (**A**) RNAscope-based in situ hybridization (ISH) revealed dense *Shh* mRNA in the GCL of P3 retina (inner nuclear layer, INL; inner plexiform layer, IPL; ganglion cell layer, GCL). (**B**) RNAscope-based ISH revealed *Shh* mRNA in CALB2[+] cells in the GCL of P3 retina. (**C**) Single-cell RNAseq data (from *Rheaume et al., 2018*) analyzed to show *Calb2* and *Shh* mRNA expression by different subtypes of RGCs in P5 mouse retina. (**D**) Single-cell RNAseq data (from *Shekhar et al., 2022*) analysis for developmental expression of *Calb2* by RGCs. (**E**) Real-time quantitative reverse transcription polymerase chain reaction (qRT-PCR) showed a reduction in *Shh* mRNA in retina of P3 *Shh^{fl/fl};Calb2^{Cre}* mice compared to controls. Each data point represents one biological replicate and bars depict means ± standard error of the mean (SEM). Asterisks (*) indicate $p < 0.05$ by Student's *t*-test (*n* = 3 mice for each group). Scale bars in A: 100 μm and in B: 10 μm.

The online version of this article includes the following figure supplement(s) for figure 3:

**Figure supplement 1.** Presence of active sonic hedgehog (SHH) signaling in developing visual thalamus.

**Figure supplement 2.** Expression of CALB2 in developing LGN and interneuron progenitor zones.

---

*2001*). To report active SHH signaling in developing visual thalamus, we tested for the presence of *Gli1*. Loss of SHH has been observed to abrogate *Gli1* expression, suggesting that it can be used as a proxy for SHH protein (*Garcia et al., 2010*). Here, staining for βGal in P3 *Gli1^{nlacZ/+}* mice demonstrated that when thalamic *Fgf15* expression is high and interneuron migration occurs, there is indeed active SHH signaling in dLGN and vLGN (*Figure 3—figure supplement 1*).

To assess the role of SHH signaling in the developing visual thalamus, we first sought approaches to delete *Ptch1* or *Smo* from astrocytes. However, we could not find an appropriate transgenic line that would achieve sufficient Cre recombination to delete these genes exclusively in astrocytes during early development. For instance, *Gfap-cre* shows only sparse Cre expression in visual thalamus before

eye opening (**Somaiya et al., 2021**). *Aldh1l1^Cre* displays Cre expression not only in astrocytes but also in some oligodendrocytes during early development (**Tien et al., 2012**) and *Aldh1l1^{Cre/ERT2}* could potentially cause significant challenges to the process of natural birth due to tamoxifen administration to pregnant mice (**Lizen et al., 2015**). Therefore, to circumvent these issues, we instead assessed the role of SHH signaling in the developing visual thalamus by crossing conditional allele *Shh^{fl/fl}* with *Calb2^Cre* mice. Multiple reasons led us to choose *Calb2^Cre* mice to target RGC-derived SHH: (1) *Shh* and *Calb2* are coexpressed in RGC subtypes (**Figure 3C**); (2) RGCs show no to very low expression of *Calb2* before E16 (**Shekhar et al., 2022**), suggesting that *Shh^{fl/fl};Calb2^Cre* mutant mice will retain other known Shh functions (**Wang et al., 2005**) in early embryonic retina (**Figure 3D**); (3) our previous studies have demonstrated significant reduction in gene expression specifically in RGCs using the *Calb2^Cre* driver line (**Kerr et al., 2019**); (4) we previously reported that CALB2^+ cells are largely absent from the neonatal and postnatal dLGN and are only sparsely distributed in vLGN at these ages (**Su et al., 2011**). Here, we add to these points by showing only a limited number of CALB2^+ cells are present in the thalamic and tectal progenitor zones that are thought to give rise to these sets of migrating interneurons (**Golding et al., 2014**; **Jager et al., 2016**; **Figure 3—figure supplement**

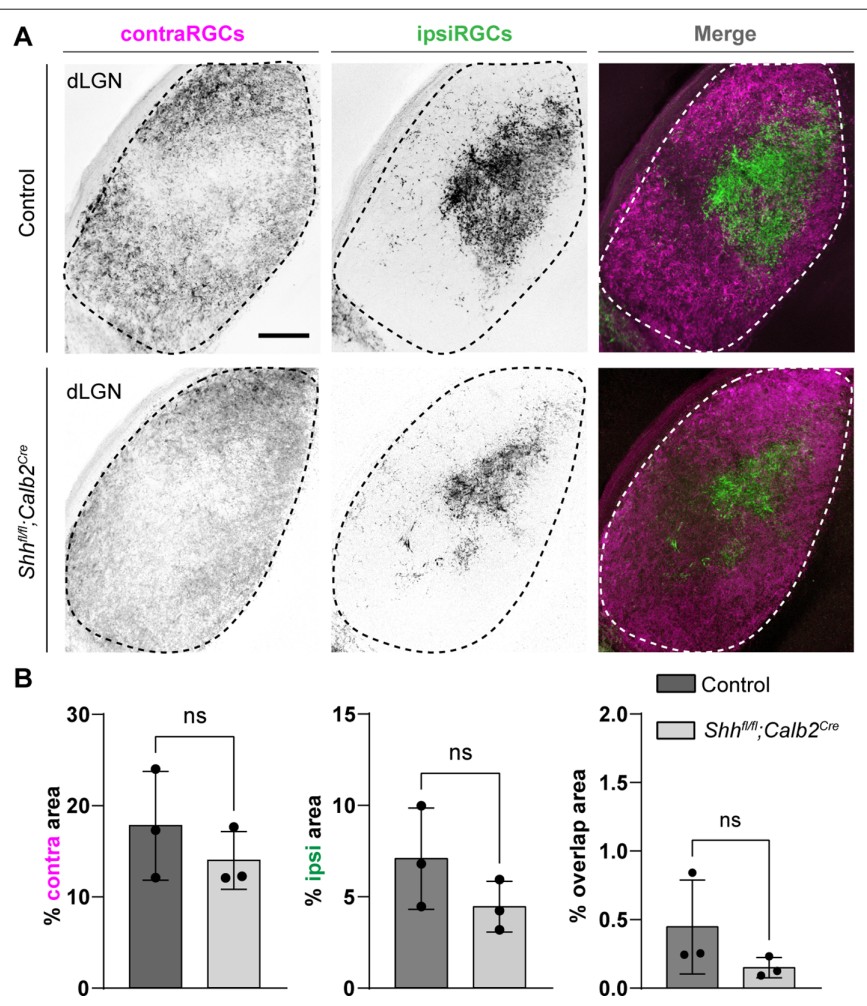

**Figure 4.** Retinal innervations innervate visual thalamus in the absence of retinal ganglion cell (RGC)-derived sonic hedgehog (SHH). (**A**) Cholera Toxin Subunit B (CTB)-labeled eye-specific retinal projections to dorsal lateral geniculate nucleus (dLGN) in P25 control and P25 *Shh^{fl/fl};Calb2^Cre* mutant mice. (**B**) Quantification for the percentage of dLGN area covered by contraRGCs projections, ipsiRGCs projections, or overlapping eye-specific projections in controls and *Shh^{fl/fl};Calb2^Cre* mutants. Each data point represents one biological replicate and bars depict means ± standard error of the mean (SEM). ns indicates no significant differences by Student's *t*-test (*n* = 3 mice for each group). Scale bar in A: 100 μm.

2). Therefore, using *Calb2^Cre* mice potentially minimizes bias caused by deletion of SHH from these regions as seen in *Shh^fl/fl Nes^Cre* mutants.

*Shh^fl/fl*;*Calb2^Cre* mutant mice are viable, fertile, and indistinguishable from littermate controls. qRT-PCR analysis confirmed a significant decrease in *Shh* mRNA in the retina of these mutants (**Figure 3E**). Intraocular CTB injections showed that RGC axons from both eyes in *Shh^fl/fl*;*Calb2^Cre* mice are capable of innervating dLGN and form normal appearing eye-specific projections in this region (**Figure 4**). The appropriate targeting of retinal axons into visual thalamus of *Shh^fl/fl*;*Calb2^Cre* mice suggests that any deficits we observe in these mutants is likely not a secondary consequence of dysinnervation in these mutants (such as the absence of any other axon-derived molecules).

Based on this reduction in RGC-derived SHH, we investigated if RGC-derived SHH is necessary for astrocytic expression of FGF15 in the developing visual thalamus. ISH analysis demonstrated a significant reduction of *Fgf15^+* cells in the dLGN and vLGN of P3 *Shh^fl/fl*;*Calb2^Cre* mutants (**Figure 5A, B**). These results resembled the reduced number of *Fgf15^+* cells in both the *Atoh7^−/−* visual thalamus and *Shh^fl/fl Nes^Cre* visual thalamus. Next, to determine if the changes in *Fgf15* expression are the result of a decrease in astrocyte numbers, we used an astrocyte-specific mRNA we previously found to be expressed by all astrocytes in the developing visual thalamus – fibroblast growth factor receptor 3 (*Fgfr3*) mRNA (**Somaiya et al., 2021**). It is important to point out for clarity that FGFR3 is not a major receptor for FGF15 (**Ornitz and Itoh, 2015**). ISH analysis showed no significant difference in the distribution of *Fgfr3*-expressing cells between the controls and mutants (**Figure 5C and D**), suggesting that loss of RGC-derived SHH does not the impact number of astrocytes in developing dLGN and vLGN. These results demonstrate that SHH signaling from RGCs is not only necessary for astrocytic expression of *Fgf15* in visual thalamus but that RGCs are the primary source of SHH to drive *Fgf15* expression in thalamic astrocytes.

Given the importance of FGF15 from our studies, we subsequently examined the effect of loss of RGC-derived SHH on thalamic interneurons. We first assessed changes in interneuron recruitment after the first week of postnatal development, at the time when these cells have fully migrated into dLGN and vLGN (**Su et al., 2020**). Our ISH data revealed that there was significant reduction in the percentage of *Gad1*-expressing cells in dLGN and vLGN of *Shh^fl/fl*;*Calb2^Cre* mice at P7 (**Figure 6A and B**). It is possible, however, that the absence of RGC-derived SHH does not completely halt thalamic interneuron recruitment, but rather delays this migration process. Thus, we also investigated changes in the number of interneurons in adult *Shh^fl/fl*;*Calb2^Cre* mice. Indeed, our data revealed a persistent, significant decrease in the percentage of *Gad1*-expressing cells in dLGN and vLGN of adult mutants (**Figure 6C and D**), similar to what we have previously reported to occur in the visual thalamus of *Fgf15^−/−* mutants (**Su et al., 2020**). Overall, our findings highlight the dependence of astrocytic *Fgf15* expression and recruitment of GABAergic interneurons into visual thalamus on the SHH signaling from retina.

## Discussion

The development of sensory neuronal circuits in many brain structures is coordinated by the arrival of sensory inputs, neuronal activity, and sensory experience (**Brunjes, 1994**; **Katz and Shatz, 1996**; **Nithianantharajah and Hannan, 2006**; **Sanes and Lichtman, 2001**). Here, we focused on the visual system where a growing body of evidence highlights critical roles for RGC axons and retinal activity in orchestrating thalamic development (**Brooks et al., 2013**; **Charalambakis et al., 2019**; **Golding et al., 2014**; **Grant et al., 2016**; **He et al., 2019**; **Sabbagh et al., 2018**; **Seabrook et al., 2013**). As one example, we previously reported that the retinal inputs signal through thalamic astrocytes to facilitate the recruitment of interneurons into dLGN and vLGN (**Su et al., 2020**). Specifically, thalamic astrocytes generate the motogen FGF15, which in contrast to other FGFs has reduced heparin-binding affinity (**Ornitz and Itoh, 2015**) making it an ideal guidance cue for long-distance migration of interneurons (**Golding et al., 2014**; **Jager et al., 2016**; **Jager et al., 2021**). In this study, we uncovered the molecular mechanism by which RGC axons signal to thalamic astrocytes to influence FGF15 expression. We show that this process is independent of RGC activity, but reliant upon SHH derived from the retina. The results highlight a SHH-dependent axo-glial–neuronal signaling mechanism important for thalamic development.

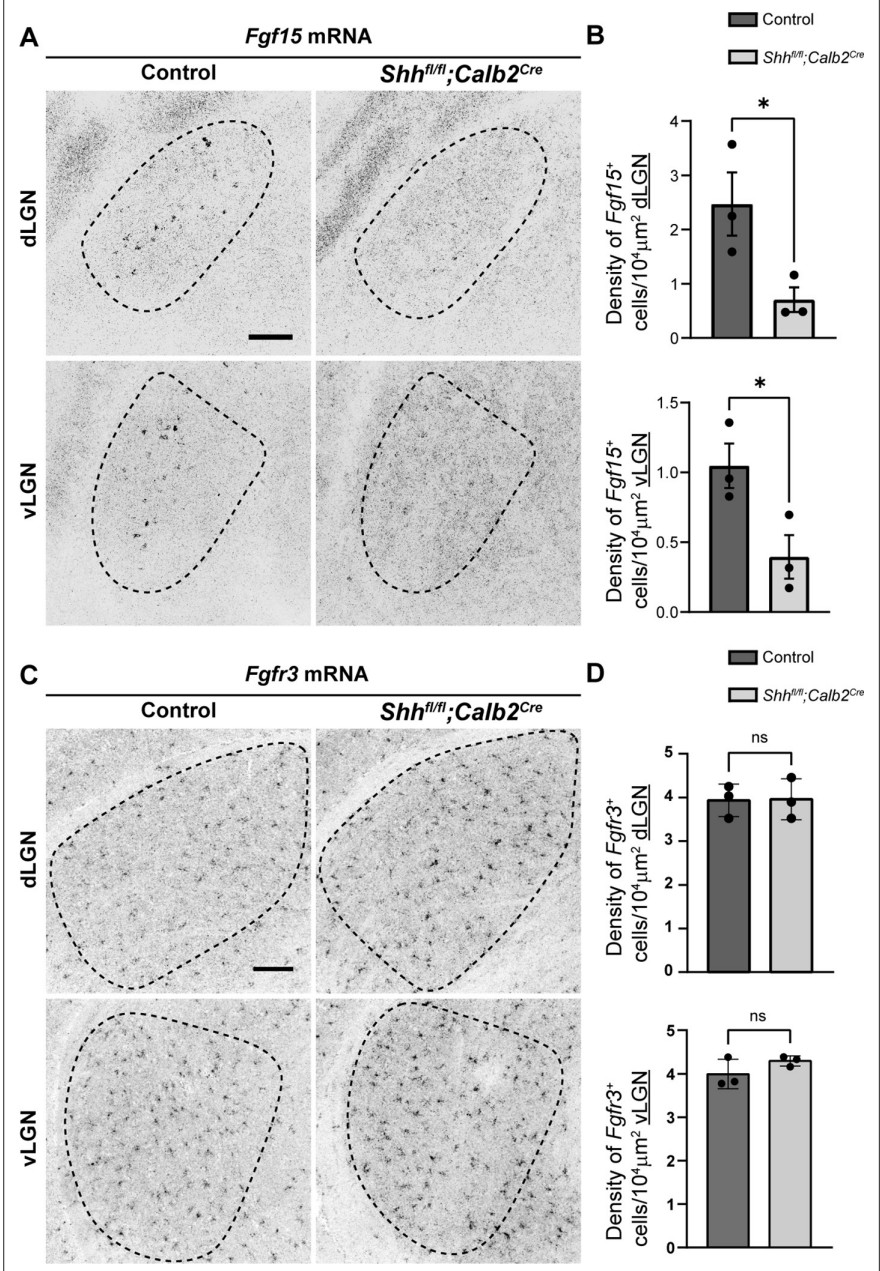

**Figure 5.** Absence of retinal ganglion cell (RGC)-derived sonic hedgehog (SHH) does not impact astrocyte distribution but decreases *Fgf15* expression in the visual thalamus. (**A**) In situ hybridization (ISH) revealed reduced *Fgf15* expression in dorsal lateral geniculate nucleus (dLGN) and ventral lateral geniculate nucleus (vLGN) of P3 *Shh^{fl/fl};Calb2^{Cre}* mutants compared to controls. (**B**) Quantification for density of *Fgf15^+* cells in dLGN and vLGN of P3 control and *Shh^{fl/fl};Calb2^{Cre}* mutant mice. Each data point represents one biological replicate and bars depict means ± standard error of the mean (SEM). Asterisks (*) indicate *p < 0.05 by Student's *t*-test (*n* = 3 mice for each group). (**C**) ISH revealed no change in *Fgfr3^+* astrocytes in dLGN and vLGN of P7 *Shh^{fl/fl};Calb2^{Cre}* mutants compared to controls. (**D**) Quantification for density of *Fgfr3^+* cells in dLGN and vLGN of P7 control and *Shh^{fl/fl};Calb2^{Cre}* mutant mice. Each data point represents one biological replicate and bars depict means ± SEM. ns indicates no significant differences by Student's *t*-test (*n* = 3 mice for each group). Scale bars in A, C: 100 µm.

## SHH as an intermediary between RGCs and thalamic astrocytes for interneuron migration

SHH signaling has well established roles in regulating morphogenesis, cell differentiation, and cell proliferation (*Garcia et al., 2018*; *Ishibashi and McMahon, 2002*; *Komada et al., 2008a*; *Marigo*

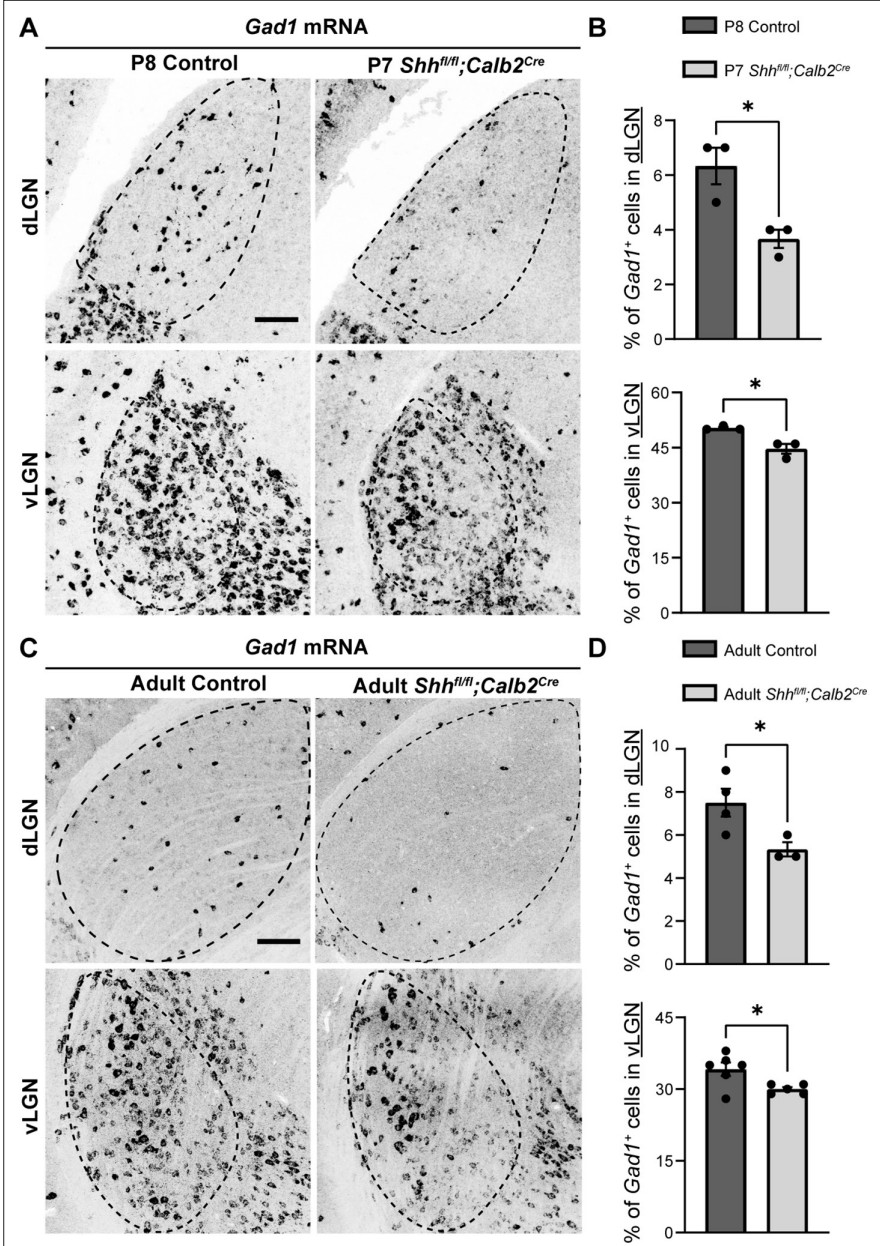

**Figure 6.** Retinal ganglion cell (RGC)-derived sonic hedgehog (SHH) is required for the recruitment of $Gad1^+$ +
into visual thalamus. (**A**) In situ hybridization (ISH) revealed a reduction in $Gad1^+$ cells in dorsal lateral geniculate
nucleus (dLGN) and ventral lateral geniculate nucleus (vLGN) of P7 $Shh^{fl/fl};Calb2^{Cre}$ mutants compared with
controls. (**B**) Quantification of percentage of $Gad1^+$ cells in dLGN and vLGN of P7 $Shh^{fl/fl};Calb2^{Cre}$ and control mice.
Each data point represents one biological replicate and bars depict means ± standard error of the mean (SEM).
Asterisks (*) indicate *p < 0.05 by Student's *t*-test ($n$ = 3 mice for each group). (**C**) ISH revealed a reduction in $Gad1^+$
cells in dLGN and vLGN of >P90 $Shh^{fl/fl};Calb2^{Cre}$ mutants compared with controls. (**D**) Quantification of percentage
of $Gad1^+$ cells in dLGN and vLGN of adult $Shh^{fl/fl};Calb2^{Cre}$ and control mice. Each data point represents one
biological replicate and bars depict means ± SEM. Asterisks (*) indicate *p < 0.05 by Student's *t*-test ($n$ = 4 mice for
control dLGN group, $n$ = 3 mice for mutant dLGN group, $n$ = 6 mice for control vLGN group, and $n$ = 5 mice for
mutant vLGN group). Scale bars in A, C: 100 μm.

*et al., 1996*). In the embryonic retina, local expression of SHH is critical for promoting and maintaining
the proliferation of retinal precursor cells (*Jensen and Wallace, 1997*; *Wang et al., 2005*). However,
here we observed that *Shh* expression is largely confined to RGCs at perinatal ages. RGC-derived
SHH protein has been shown to be anterogradely transported along axons, where it maintains the

proliferative capacity of astrocytes in the optic nerve (*Dakubo et al., 2008*; *Wallace and Raff, 1999*) and acts as cue to guide eye-specific axon segregation at the optic chiasm (*Peng et al., 2018*). Radiolabeling experiments have reported the presence of retina-derived SHH as far as the superior colliculus (*Traiffort et al., 2001*), a midbrain retinorecipient region, suggesting previously unexplored roles for RGC-derived SHH likely exist beyond the optic nerve and chiasm.

In the current study, we discovered that retinal-derived SHH plays important roles in the recruitment of interneurons into visual thalamus, a major RGC target site in the brain. These results are in line with several recent studies with showing that neuron-derived SHH directly acts on astrocytes residing in the target regions of their axonal projections (*Garcia et al., 2010*; *Hill et al., 2019*). We show that astrocytes in visual thalamus generate several downstream components of SHH signaling and that RGC-derived SHH is critical for astrocytic *Fgf15* expression in visual thalamus. While FGF15 expression has been shown to be dependent on local SHH signaling in the embryonic brain (*Ishibashi and McMahon, 2002*; *Kim et al., 2018*; *Komada et al., 2008b*; *Saitsu et al., 2005*; *Yabut et al., 2019*), this is the first study to provide evidence for long-distance SHH signaling to regulate the expression of astrocyte-derived FGF15.

The loss of astrocyte-derived FGF15 in $Shh^{fl/fl};Calb2^{Cre}$ mutants results in a significant loss of $Gad1^+$ interneurons in both dLGN and vLGN. However, it does not result in an absence of these GABAergic neurons. In other brain regions, the loss of SHH signaling has been observed to dramatically affect specific subtypes of GABAergic neurons, such as interneurons that express parvalbumin (Pvalb) or Somatostatin (Sst) (*Ihrie et al., 2011*; *Xu et al., 2005*), which may explain the partial loss of $Gad1^+$ cells in the visual thalamus of $Shh^{fl/fl};Calb2^{Cre}$ mutants. In dLGN, transcriptional, morphological, and functional studies have been unable to parse interneurons into more than one or two subtypes (*Charalambakis et al., 2019*; *Leist et al., 2016*), making it difficult to study the specificity for loss of $Gad1^+$ cells in this retinorecipient brain region. In contrast, vLGN contains numerous transcriptionally distinct GABAergic cell types (including both local interneurons and projection neurons), some of which exhibit regional preferences, unique morphologies, and distinct connectivity (*Fratzl et al., 2021*; *Sabbagh et al., 2021*; *Salay and Huberman, 2021*). Thus, RGC-derived SHH may affect specific subtypes of GABAergic cells in vLGN. Unfortunately, we currently lack transcriptional or neurochemical markers to specifically label interneurons in vLGN (vs. GABAergic projection neurons) making this challenging to address at this time.

It is also noteworthy that we observed more dramatic decrease in $Gad1^+$ cells in $Shh^{fl/fl};Nes^{Cre}$ mutants, particularly in the vLGN, compared to the reduction observed in either $Shh^{fl/fl};Calb2^{Cre}$ or $Atoh7^{-/-}$ mice. We interpret this to indicate multiple roles and sources of SHH contribute to thalamic development. In $Shh^{fl/fl};Nes^{Cre}$ mutants, SHH loss occurs much earlier in development and in progenitor zones (*Lendahl et al., 1990*; *Zimmerman et al., 1994*), likely impacting the generation of GABAergic progenitors that will eventually migrate into visual thalamus. Indeed, *Shh* null mice show defects in ventral patterning and a noticeable decrease in the size of their brain (*Chiang et al., 1996*; *Machold et al., 2003*; *Rallu et al., 2002*), indicating key roles for this morphogen in many aspects of early brain development. In $Shh^{fl/fl};Calb2^{Cre}$ mice, SHH is not deleted from many cells in these progenitor zones, suggesting to us that these results differentiate between progenitor zone-derived SHH functions and RGC-derived SHH functions. An alternative possibility, however, is that SHH is also generated by cells within visual thalamus, which could also potentially influence interneuron migration process. While we detect no to very low expression of *Shh* in neonatal visual thalamus with microarray and sequencing techniques, this could reflect technical limitation in these approaches or analysis at the wrong ages (*Monavarfeshani et al., 2018*; *Su et al., 2011*).

## SHH signaling in RGC subtypes

Even though most RGC subtypes produce SHH, one fascinating question is whether there are any functional differences between those that express high versus low levels of this molecule. Clusters 16, 19, and 32 were notable for showing high SHH expression (*Figure 3C*). The transcription factor *Pou6f2* reported to specify these RGC types (*Rheaume et al., 2018*) is expressed in a novel class of On–Off direction selective RGCs (DSGCs) (*Li et al., 2020*). Conversely, RGC clusters 30 and 40 that are defined by transcription factor *Satb2* (*Rheaume et al., 2018*) showed very low or no expression of SHH (*Figure 3C*). *Satb2* is a selective marker for On–Off DSGCs encoding either posterior or anterior motion (*Dhande et al., 2019*). Why do DSGCs subtypes exhibit varying degrees of SHH expression?

One possibility is that each DSGC subtypes contributes differently to interneuron development, some of which are independent of SHH. Indeed, retinal inputs are not only important for interneuron migration but also their absence leads to arrested arbor branching and dystrophic arbor field growth of these GABAergic cells (*Charalambakis et al., 2019*). Studies examining interneuron development under specific RGC subtype manipulation may provide further insight into these differences.

## SHH signaling in astrocytes

While SHH signaling is well characterized for its dynamic roles in neural precursor cells and oligodendrocytes (*Belgacem et al., 2016*), much less is known about its function in astrocyte development, particularly in the context of the developing visual system. In the optic nerve, SHH has been reported to be necessary for maintenance of astrocyte proliferation (*Dakubo et al., 2008*; *Wallace and Raff, 1999*). Our work demonstrates that this signaling pathway is also critical for the expression of *Fgf15* by thalamic astrocytes. However, only a small fraction of astrocytes express *Fgf15* in visual thalamus (*Su et al., 2020*), raising the questions of whether only some thalamic astrocytes generate the machinery to respond to SHH or whether retina-derived SHH itself can alter the identity of thalamic astrocytes to generate heterogeneity in these cells. In support of the latter possibility, neuron-derived SHH is integral for maintaining the identity of Bergmann glial cells, a specialized astrocyte type in the cerebellum (*Farmer et al., 2016*). However, it is important to note that we found that a much greater proportion of astrocytes in visual thalamus generate SHH signaling components, *Ptch1* and *Smo*, compared to those that generate *Fgf15*. One possibility is that all thalamic astrocytes can generate SHH-induced *Fgf15*, but they do it asynchronously. The widespread expression of SHH signaling components in thalamic astrocytes also suggests that this pathway could be important in ways beyond interneuron migration. Outside of the thalamus, SHH appears to be important for a variety of astrocytic functions, including modulation of neuronal activity (*Hill et al., 2019*), regulation of energy metabolism (*Tirou et al., 2021*), and neuroprotection (*Ugbode et al., 2017*). These studies highlight the need to further explore the roles for SHH signaling in thalamic circuits.

## Neuronal activity from RGCs and thalamic interneuron migration

Many developmental aspects underlying orderly connections in the mammalian visual system are dependent on retinal activity even before eye opening (*D'Souza and Lang, 2020*; *Huberman et al., 2008*). In the case of thalamic interneurons, the role of retinal activity in their migration, differentiation, or incorporation into thalamic circuits was unresolved prior to these studies. Application of tetrodotoxin in neonatal organotypic thalamic slices led to the suggestion that activity was necessary for directing the spatial incorporation of these interneurons into dLGN circuits (*Golding et al., 2014*). In the present study, we investigated whether in vivo RGC activity contributes to the initial recruitment of GABAergic interneurons into dLGN and vLGN. Our results revealed that decreasing retinal activity had little effect on the percentage of interneurons recruited into dLGN and vLGN. Although this genetic approach to inhibit neuronal activity has been shown to be effective in the developing hippocampus (*Sando et al., 2017*), it is possible that we may not have completely blocked the release of neurotransmitters from RGCs in $Rosa26^{floxstop-TeNT}$;$Calb2^{Cre}$ mutants (because of incomplete expression of TeNT by all RGCs, developmentally regulated expression of TeNT or other confounding issues) and that the remaining activity was sufficient to directly influence the interneuron migration process. In fact, it is possible that retinal activity may even induce RGCs to release SHH. Although such roles of neuronal stimulation on SHH release have not been reported in visual thalamus, activity in hippocampal neurons has been shown to cause the release of SHH in a SNAREs- and $Ca^{2+}$-dependent manner (*Su et al., 2017*). We argue that is not the case based on the early and broad expression of Cre in the driver line applied here – ensuring all subtypes of RGCs will express TeNT at birth. Moreover, the impaired refinement of eye-specific RGC projections in $Rosa26^{floxstop-TeNT}$;$Calb2^{Cre}$ mice similar to impairments induced by manipulation and blocking of retinal activity early in perinatal development (*Huberman et al., 2003*; *Pfeiffenberger et al., 2005*) suggests that activity is functionally reduced in these mutants. Overall, this suggests that retinal inputs have both activity-dependent and activity-independent roles in orchestrating the development of visual thalamus. The techniques typically used to study how retinal inputs impact the thalamic development, surgical or genetic removal of RGC axons from thalamus have failed to distinguish between these independent roles previously.

# Materials and methods

### Key resources table

| Reagent type (species) or resource | Designation | Source or reference | Identifiers | Additional information |
|---|---|---|---|---|
| Antibody | anti-GFP (rabbit polyclonal) | Thermo Fisher Scientific | Cat#A-11122; RRID:AB_221569 | 1:250 |
| Antibody | anti-S100 (rabbit polyclonal) | Dako | Cat# Z0311; RRID:AB_10013383 | 1:200 |
| Antibody | anti-Calretinin (rabbit polyclonal) | Swant | Cat#7697; RRID: AB_2619710 | 1:1000 |
| Antibody | anti-RFP (rabbit polyclonal) | Thermo Fisher Scientific | Cat#600-401-379-RTU; RRID:AB_2209751 | 1:500 |
| Antibody | Anti-Digoxigenin-POD (sheep polyclonal) | Millipore Sigma | Cat#11207733910; RRID:AB_514500 | 1:1000 |
| Antibody | Anti-Fluorescein-POD (sheep polyclonal) | Millipore Sigma | Cat#11426346910; RRID:AB_840257 | 1:1000 |
| Biological sample (*Mus musculus*) | *Rosa26^{tdT};Gli1^{CreER}* brains | A.D.R. Garcia, Drexel University | JAX #007913, #007914; RRID: IMSR_JAX:007913, IMSR_JAX:007914 | |
| Peptide, recombinant protein | Fluorescein RNA Labeling Mix | Roche | Cat#11685619910 | |
| Peptide, recombinant protein | DIG RNA Labeling Mix | Roche | Cat#11277073910 | |
| Peptide, recombinant protein | SuperScript II Reverse Transcriptase | Thermo Fisher Scientific | Cat#18064022 | |
| Peptide, recombinant protein | Cholera Toxin Subunit B (CTB, Recombinant), Alexa Fluor 488 Conjugate | Thermo Fisher Scientific | CAT#C22841 | |
| Peptide, recombinant protein | Tamoxifen | Sigma | CAT#T5648-1G | |
| Peptide, recombinant protein | CTB (Recombinant), Alexa Fluor 555 Conjugate | Thermo Fisher Scientific | CAT#C34776 | |
| Commercial assay, kit | SuperScript II Reverse Transcriptase First Strand cDNA Synthesis kit | Invitrogen | Cat#18064014 | |
| Commercial assay, kit | pGEM-T Easy Vector Systems | Promega | Cat#A1360 | |
| Commercial assay, kit | MAXIscript in vitro Transcription Kit | Ambion | Cat#AM1312 | |
| Commercial assay, kit | Tyramide Signal Amplification system | PerkinElmer | Cat#NEL753001KT | |
| Commercial assay, kit | iTaq Universal SYBR Green Supermix | Bio-Rad | Cat#1725124 | |
| Commercial assay, kit | Bio-Rad Total RNA Extraction from Fibrous and Fatty Tissue kit | Bio-Rad | Cat#7326870 | |
| Commercial assay, kit | RNAscope Multiplex Fluorescent Reagent Kit V2 | Advanced Cell Diagnostics (ACD) | Cat#323100 | |
| Other | RNAseq datasets for the developing mouse dLGN and vLGN | DOI: https://doi.org/10.7554/eLife.33498.006 | | *Monavarfeshani et al., 2018* |
| Other | Single-cell RNAseq dataset for RGC subtypes | DOI: https://doi.org/10.1038/s41467-018-05134-3 | Accession # GSE115404 | *Rheaume et al., 2018* |
| Other | Single-cell RNAseq dataset for RGCs at various ages | DOI: https://doi.org/10.7554/eLife.73809 | Accession # GSE185671 | *Shekhar et al., 2022* |

*Continued on next page*

*Continued*

| Reagent type (species) or resource | Designation | Source or reference | Identifiers | Additional information |
|---|---|---|---|---|
| Strain, strain background (*Mus musculus*) | C57BL/6J mice | The Jackson Laboratory | JAX#000664; RRID:IMSR_JAX:000664 | |
| Strain, strain background (*Mus musculus*) | Calb2Cre | The Jackson Laboratory | JAX#010774; RRID:IMSR_JAX:010774 | |
| Strain, strain background (*Mus musculus*) | Shhfl/fl | The Jackson Laboratory | JAX#004293; RRID:IMSR_JAX:004293 | |
| Strain, strain background (*Mus musculus*) | NesCre | The Jackson Laboratory | JAX#003771; RRID:IMSR_JAX:003771 | |
| Strain, strain background (*Mus musculus*) | Aldh1l1-GFP | S. Robel, Virginia Tech | Stock#011015-UCD; RRID: MMRRC_011015-UCD | |
| Strain, strain background (*Mus musculus*) | Rosa26floxstop-TeNT | A. Maximov, The Scripps Research Institute | MGI:3839913 | *Zhang et al., 2008* |
| Strain, strain background (*Mus musculus*) | Rosa26tdT(Ai14) | The Jackson Laboratory | JAX#007914; RRID: IMSR_JAX:007914 | |
| Strain, strain background (*Mus musculus*) | Gli1CreER | *Ahn and Joyner, 2005* | JAX#007913; RRID: IMSR_JAX:007913 | |
| Strain, strain background (*Mus musculus*) | Rosa26tdT | The Jackson Laboratory | JAX#007909; RRID:IMSR_JAX:007909 | |
| Strain, strain background (*Mus musculus*) | Atoh7−/− | S.W. Wang, University of Texas MD Anderson Cancer Center | Stock# 042298-UCD; RRID:MMRRC_042298-UCD | |
| Strain, strain background (*Mus musculus*) | Gli1nlacZ/+ | The Jackson Laboratory | JAX#008211; RRID:IMSR_JAX:008211 | *Bai et al., 2002* |
| Sequence-based reagent | Gad1 cloning primer F: TGTGCCCAAACTGGTCCT; R: TGGCCGATGATTCTGGTT | Integrated DNA Technologies | N/A | |
| Sequence-based reagent | Gja1 cloning primer F: CGTGAAGGGAAGAAGCGA; R: GCCTGCAAACTGCCAAGT | Integrated DNA Technologies | N/A | |
| Sequence-based reagent | Shh qPCR primer F: ACGTAGCCGAGAAGACCCTA; R: ACTTGTCTTTGCACCTCTGAGT | Integrated DNA Technologies | N/A | |
| Sequence-based reagent | Gapdh qPCR primer F: CGTCCCGTAGACAAAATGGT; R: TTGATGGCAACAATCTCCAC | Integrated DNA Technologies | N/A | |
| Sequence-based reagent | 18s qPCR primer F: GGACCAGAGCGAAAGCATTTG; R: GCCAGTCGGCATCGTTTATG | Integrated DNA Technologies | N/A | |
| Sequence-based reagent | Cre genotyping primer F: CGTACTGACGGTGGGAGAAT; R: TGCATGATCTCCGGTATTGA | Integrated DNA Technologies | N/A | |
| Sequence-based reagent | Shhfl/fl genotyping primer F: CAGAGAGCATTGTGGAATGG; R: CAGACCCTTCTGCTCATGG | Integrated DNA Technologies | N/A | |

*Continued on next page*

*Continued*

| Reagent type (species) or resource | Designation | Source or reference | Identifiers | Additional information |
|---|---|---|---|---|
| Sequence-based reagent | tdT genotyping primer F: ACCTGGTGGAGT TCAAGACCATCT; R: TTGATGACGGCCA TGTTGTTGTCC | Integrated DNA Technologies | N/A | |
| Sequence-based reagent | GFP genotyping primer F: AAGTTCATCTGCACCACCG; R: TCCTTGAAGAAGATGGTGCG | Integrated DNA Technologies | N/A | |
| Sequence-based reagent | TeNT genotyping primer FA: AAAGTCGCTCTGAGTTGTTAT; RA: GGAGCGGGAGAAATGGATATG; SA: CATCAAGGAAACCC TGGACTACTG | Integrated DNA Technologies | N/A | |
| Sequence-based reagent | *Atoh7*$^{-/-}$ genotyping primer (to see the wild-type band) F: ATGGCGCTCAGCTACATCAT; R: GGGTCTACCTGGAGCCTAGC | Integrated DNA Technologies | N/A | |
| Sequence-based reagent | *Neo* genotyping primer (to see the mutant *Atoh7* band) F: GCCGGCCACAGTCGATGAATC; R: CATTGAACAAGATGGATTGCA | Integrated DNA Technologies | N/A | |
| Recombinant DNA reagent | Mouse Fgf15 cDNA | Horizon (Dharmacon) | Cat#MMM1013-202768318, Clone ID: 5066286 | |
| Recombinant DNA reagent | RNA scope probe-Mm-Smo | ACD | Cat#318411 | |
| Recombinant DNA reagent | RNA scope probe-Mm-Ptch1-C2 | ACD | Cat#402811-C2 | |
| Recombinant DNA reagent | RNA scope probe-Mm-Shh-C3 | ACD | Cat#314361-C3 | |
| Recombinant DNA reagent | RNA scope 3-plex positive control probe-mm | ACD | Cat#320881 | |
| Recombinant DNA reagent | RNA scope 3-plex negative control probe-mm | ACD | Cat#320871 | |
| Software, algorithm | Prism | GraphPad | Version 8.0; RRID: SCR_002798 | |
| Software, algorithm | Adobe Photoshop | Adobe Inc | Version: 21.1.2 | |
| Software, algorithm | ZEN black edition | Carl Zeiss | Version: 14.0.12.201 | |
| Software, algorithm | Fiji ImageJ | NIH | Version: 1.52p | |
| Software, algorithm | RStudio | RStudio, Inc | Version: 1.2.5042 | |
| Other | *Fgf15* riboprobe | This paper | N/A | Information in 'Riboprobe production' |
| Other | *Gad1* riboprobe | This paper | N/A | Information in 'Riboprobe production' |
| Other | *Gja1* riboprobe | This paper | N/A | Information in 'Riboprobe production' |

## Animals

C57BL/6J, *Calb2*$^{Cre}$, *Shh*$^{fl/fl}$, *Nes*$^{Cre}$, and *Rosa26*$^{tdT}$ mice were obtained from The Jackson Laboratory. *Atoh7*$^{-/-}$ mice were obtained from S.W. Wang (University of Texas MD Anderson Cancer Center, Houston, TX). *Aldh1l1-GFP* mice were provided by S. Robel (Virginia Tech, Roanoke, VA). *Rosa26*$^{floxstop-TeNT}$ mice were obtained from A. Maximov (The Scripps Research Institute, La Jolla, CA), after receiving approval from M. Goulding (The Salk Institute for Biological Studies, La Jolla, CA). Tissue from *Gli1*$^{nlacZ/+}$ mice was obtained from A.D.R. Garcia (Drexel University, Philadelphia, PA). We were unable to breed *Rosa26*$^{floxstop-TeNT}$;*Calb2*$^{Cre}$ with homozygous TeNT, possibly due to embryonic death. Thus, for any activity-related experiment presented in *Figure 1* and *Figure 1—figure supplement 2*, mice

heterozygous for the TeNT allele were considered as mutant mice. The key resources table includes sequences for the genotyping primers. *Gli1*$^{CreER}$ mice (*Ahn and Joyner, 2005*) were crossed with Ai14 reporter mice (JAX #007914) to generate *Rosa26*$^{tdT}$;*Gli1*$^{CreER}$ mice.

## Preparation of tissue and IHC

As previously described, tribromoethanol (Avertin) was intraperitoneally injected into mice at a concentration of 12.5 µg/ml. The animals were trans-cardially perfused with phosphate-buffered saline (PBS) and 4% paraformaldehyde (PFA, pH 7.4) (*Su et al., 2010*). Brains were kept overnight at 4°C in 4% PFA, and then transferred to 30% sucrose in PBS for at least 48 hr. The fixed brains were embedded in Tissue Freezing Medium (Electron Microscopy Sciences) and 16 µm thick sections were cryosectioned on a Leica CM1850 cryostat. Following air-drying for 30 min, slides were incubated in IHC blocking buffer (2.5% bovine serum albumin, 5% normal goat serum, 0.1% Triton-X in PBS) for 30 min. Primary antibodies were diluted in blocking buffer and incubated on the sections at 4°C for >18 hr (information on the antibodies is available in the key resources table). Following washing with PBS, fluorophore-conjugated secondary antibodies (Invitrogen) were incubated on tissue sections for 1 hr at room temperature. The sections were stained with 4',6-diamidino-2-phenylindole (DAPI) after several washes in PBS and mounted with Vectashield (Vector Laboratories). Bright-field βGal staining was performed as previously described staining (*Garcia et al., 2010*).

## Riboprobe production

Riboprobes were generated as described previously (*Monavarfeshani et al., 2018*; *Su et al., 2010*). Plasmids carrying *Fgf15* were purchased from GE Dharmacon. *Gad1* 1 kb cDNA (corresponding to nucleotides 1099–2081) and *Gja1* 1.1 kb cDNA (corresponding to nucleotides 714–1854) were generated using SuperScript II Reverse Transcriptase First Strand cDNA Synthesis kit and the manufacturer's protocol. The information for the sequence of cloning primers is in the key resources table. cDNA was then cloned into the pGEM-T Easy vector. We generated sense and antisense riboprobes against *Fgf15*, *Gad1*, and *Gja1* from linearized plasmids using DIG- or FL-labeled uridylyltransferase and MAXIscript in vitro Transcription Kit. Riboprobes were hydrolyzed into 500 bp fragments by adding 6 µl of Na$_2$CO$_3$ (1 M), 4 µl of NaHCO$_3$ (1 M), and 80 µl of water at 60°C. Following precipitation with ethanol, the riboprobes were dissolved in RNAase-free water.

## Tamoxifen administration

Tamoxifen was dissolved in corn oil at 5 mg/ml. Tamoxifen was administered by intragastric injection to *Rosa26*$^{tdT}$;*Gli1*$^{CreER}$ pups at P0.

## In situ hybridization (ISH) with in-house generated riboprobes

We performed ISH using the generated riboprobes on PFA-fixed, cryosectioned 16 µm tissue as described previously (*Monavarfeshani et al., 2018*). The sections were fixed in 4% PFA for 10 min, washed with PBS, and incubated with proteinase K (1 µg/ml in 50 mM Tris pH 7.5, 5 mM ethylenediaminetetraacetic acid (EDTA)) solution for 10 min. After being washed with PBS, sections were incubated for 5 min in 4% PFA, washed with PBS, and then incubated for 10 min in acetylation solution (0.25% acetic anhydride, 20 mM hydrochloric acid, and 1.33% triethanolamine). In order to permeabilize them, sections were incubated for 30 min in 1% Triton in PBS. For blocking endogenous peroxidase, sections were incubated in 0.3% hydrogen peroxide in PBS for 30 min, then rinsed in PBS. We equilibrated the sections in hybridization solution (50 ml of prehyb solution, 1.6 ml of 5 mg/ml, and 25 mg Roche yeast RNA) for 1 hr, and then incubated them in heat-denatured diluted riboprobes (10 min at 80°C) overnight at 65°C. On the next day, slides were rinsed with 0.2× saline-sodium citrate solution followed by Tris-buffered saline (TBS). Following 1-hr incubation in blocking buffer (10% lamb serum, 0.2% Roche blocking reagent in TBS), the slides were incubated overnight at 4°C in horseradish peroxidase(HRP)-conjugated anti-DIG or anti-FL antibodies. On day 3, riboprobes were identified using a Tyramide Signal Amplification (TSA) system.

## ISH with commercially generated RNAscope riboprobes

The probes used were RNA scope probe-Mm-Smo, RNA scope probe-Mm-Ptch1-C2, RNA scope probe-Mm-Shh-C3, RNA scope 3-plex positive control probe-mm, and RNA scope 3-plex negative

control probe-mm. PFA-fixed, cryosectioned 16 µm tissue were processed for RNA Scope Multiplex using the manufacturer's protocol. Briefly, sections were dehydrated by ethanol treatment and then pretreated with target retrieval reagent and protease III. Following the hybridization of probes, we amplified signals on sections using the TSA system.

### Intraocular injection of CTB

We anesthetized mice with isoflurane or hypothermia for intraocular injections, as described previously (*Jaubert-Miazza et al., 2005*). A fine glass pipette attached to a picospritzer was used to inject 1–2 µl of CTB, 1 mg/ml, intravitreally into the eye. After 2–3 days, mice were perfused, and their PFA-fixed brains were sectioned (100 µm) using a Vibratome (HM650v, Thermo Fisher Scientific) and mounted with Vectashield.

### Imaging

A Zeiss LSM700 confocal microscope was used for image acquisition. Each representative image in the figure is a maximum intensity projection. Colocalization was confirmed using single-plane images.

### qRT-PCR

RNA was isolated from non-pooled P3 control and *Shh^{fl/fl};Calb2^{Cre}* mutant retina using the Bio-Rad Total RNA Extraction from Fibrous and Fatty Tissue kit and manufacturer's protocol. Using this RNA, cDNAs were generared with SuperScript II Reverse Transcriptase. qPCR was performed using iTaq Universal SYBR Green Supermix on a CFX Connect Real-Time system (Bio-Rad). Cyling conditions for 500 or 1000 ng of cDNA are as follows: 95°C for 30 s and 42 cycles of amplification (95°C for 10 s, 60°C for 30 s) followed by a melting curve analysis. Each individual run included separate 18S rRNA or glyceraldehyde-3-phosphate dehydrogenase (*gapdh*) control reactions. Using the $2^{-\Delta\Delta CT}$ method, we determined the relative quantities of RNA (*Schmittgen and Livak, 2008*). The primer list can be found in the key resources table.

### Looming visual stimulus assay

The looming visual stimulus assay was performed as described previously (*Su et al., 2021*). Mice were presented with an overhead looming stimulus in a white rectangular arena (47 × 37 × 30 cm). To maintain uniform environmental conditions, the arena was illuminated evenly from above and enclosed in a light-proof, sound-isolating room. One of the corners had an opaque shelter, whose entrance faced the center of the arena. On the stand next to the arena, a camera was mounted to capture mouse behavior. The mice were only tested once each to prevent habituation to the stimulus. Ten minutes prior to the recording of the test, the animals were free to explore the arena and the shelter. Looming stimulus began when the animal moved around the center of the arena. Ten seconds of video was recorded prior to, during, and after the stimulus onset. We manually scored the animal's behavior during 10 s of looming stimulus. The reaction time was defined as a period of one or more seconds in which the animal either freezes, runs, or hides in the shelter after the stimulus began.

### Quantification and statistical analysis

A minimum of three biological replicates per genotype and age and a minimum of three dLGN/vLGN sections per animal were used for all quantification, based on observed variability and prior experience (*Sabbagh et al., 2018*; *Sabbagh et al., 2021*; *Su et al., 2020*). We did not exclude any data or animals from our analyses. No sex-specific differences were observed. Statistical analyses (Student's *t*-test) were performed using GraphPad Prism (version 8.0). $p < 0.05$ values were considered to be significantly different. The figure legends provide p values for all experiments. Data are plotted as mean ± standard error of the mean.

### Percentage of *Gad1^+*, *Gja1^+*, and *Gli1-tdT^+* cells

Images were obtained using ×10 magnification. We quantified *Gad1^+*, *Gja1^+* (by ISH), and *Gli1-tdT^+* (by *Rosa26^{tdT};Gli1^{CreER}* reporter mice) cells and divided by DAPI^+ cells counted in that tissue section. 'Count Tool' function in Adobe Photoshop (version: 21.1.2) was utilized for counting purposes.

## Density of *Fgf15⁺* cells

Images were obtained using ×10 magnification. We quantified *Fgf15⁺* (by ISH) cells and normalized them to the area of dLGN or vLGN. Areas were measured by outlining the boundaries of dLGN or vLGN using ZEN 2.3 SP1 FP1 (black) edition (version: 14.0.12.201, Carl Zeiss).

## CTB analysis

Images were obtained using ×10 magnification and Fiji ImageJ (version: 1.52p, NIH) was used for analyses. Using 'Split Channels', CTB 488 and CTB 555 signal were separated. For each channel, contrast was enhanced ('Enhance Contrast', 0.3%), background was subtracted ('Subtract Background', 1000 pixels and 'Math', 'Subtract'), and channels were binarized ('Make Binary'). The dLGN boundary was drawn manually by utilizing the 'Freehand selections' tool. Using this boundary and 'Area fraction' on each binarized channel, percentage of dLGN covered by ipsiRGCs or contraRGCs was determined. To obtain the overlap channel, we used 'AND' function in 'Image Calculator'. On this channel, the dLGN boundary was drawn and 'Area fraction' gave the percentage of dLGN area covered by the overlap of ipsi and contraRGCs projections.

## RNAscope analysis

Images were acquired at ×20 magnification. We used ACD's scoring criteria for this analysis (https://acdbio.com/dataanalysisguide). A cell was classified as positive for *Ptch1* or *Smo* (by RNAscope) if it contained at least 10 dots. This was divided by *Aldh1l1-GFP⁺* cells (by *Aldh1l1-GFP* transgenic line) to obtain percentage of astrocytes that express *Ptch1* or *Smo* in dLGN and vLGN. 'Count Tool' function in Adobe Photoshop was utilized for counting purposes.

## *Calb2* and *Shh* expression in RGC clusters

Single-cell data from *Rheaume et al., 2018* were downloaded from NCBI GEO (accession #: GSE115404). *Shh* (Ensembl ID = ENSMUSG00000002633) and *Calb2* (Ensemble ID = ENSMUSG00000003657) expression values were used to generate 'ggplot2' on RStudio (version: 1.2.5042, RStudio, Inc).

## Acknowledgements

This work was supported by National Institutes of Health grants EY021222 (MAF), EY030568 (MAF), EY033528 (MAF and JNC), HL153916 (JNC), and American Diabetes Association Pathway to Stop Diabetes Award 1-18-INI-14 (JNC). We are grateful to the members of the MAF lab for scientific discussion and comments on the manuscript. We thank Dr. Stefanie Robel for providing *Aldh1l1-EGFP* mice, Dr. Steven W Wang for providing *Atoh7⁻/⁻* mice, and Dr. Anton Maximov for providing *Rosa26*ˡᵒˣˢᵗᵒᵖ⁻ᵀᵉᴺᵀ mice. The authors also thank Dr. Karthik Shekhar (University of California, Berkeley) for input on bioinformatics approaches.

## Additional information

### Funding

| Funder | Grant reference number | Author |
| --- | --- | --- |
| National Eye Institute | EY021222 | Michael A Fox |
| National Eye Institute | EY030568 | Michael A Fox |
| National Eye Institute | EY033528 | John N Campbell |
| National Heart, Lung, and Blood Institute | HL153916 | John N Campbell |

The funders had no role in study design, data collection, and interpretation, or the decision to submit the work for publication.

## Author contributions
Rachana Deven Somaiya, Conceptualization, Formal analysis, Investigation, Methodology, Writing – original draft; Katelyn Stebbins, Formal analysis, Methodology, Writing – review and editing; Ellen C Gingrich, Formal analysis, Investigation; Hehuang Xie, Data curation, Formal analysis, Writing – review and editing; John N Campbell, Data curation, Formal analysis, Investigation; A Denise R Garcia, Resources, Methodology, Writing – review and editing; Michael A Fox, Conceptualization, Supervision, Funding acquisition, Methodology, Project administration, Writing – review and editing

## Author ORCIDs
Rachana Deven Somaiya ⬡ http://orcid.org/0000-0002-1190-1192
A Denise R Garcia ⬡ http://orcid.org/0000-0001-5809-3543
Michael A Fox ⬡ http://orcid.org/0000-0002-1649-7782

## Ethics
C57BL/6J, Calb2Cre, Shhfl/fl, NesCre, and Rosa26tdT mice were obtained from The Jackson Laboratory. Atoh7−/− mice were obtained from S. W. Wang (University of Texas MD Anderson Cancer Center, Houston, TX). Aldh1l1-GFP mice were provided by S. Robel (Virginia Tech, Roanoke, VA). Rosa26floxstop-TeNT mice were obtained from A. Maximov (The Scripps Research Institute, La Jolla, CA), after receiving approval from M. Goulding (The Salk Institute for Biological Studies, La Jolla, CA). Tissue from Gli1nlacZ/+ mice was obtained from A. D. R. Garcia (Drexel University, Philadelphia, PA). We were unable to breed Rosa26floxstop-TeNT;Calb2Cre with homozygous TeNT, possibly due to embryonic death. Thus, for any activity-related experiment presented in Figure 1 and Figure 1—figure supplement 2, mice heterozygous for the TeNT allele were considered as mutant mice. The key resources table includes sequences for the genotyping primers. Gli1CreER mice (Ahn and Joyner, 2005) were crossed with Ai14 reporter mice (JAX #007914) to generate Rosa26tdT;Gli1CreER mice. Ethics: Both sexes were used for all experiments. Animals had ad libitum access to water and food and were housed in a temperature-controlled environment and in a 12-hr light/dark cycle. The experiments were approved by the Virginia Tech Institutional Animal Care and Use Committee (VT Protocols 21-085 and 21-130).

## Decision letter and Author response
Decision letter https://doi.org/10.7554/eLife.79833.sa1
Author response https://doi.org/10.7554/eLife.79833.sa2

# Additional files

## Supplementary files
• MDAR checklist

## Data availability
No new large sequencing datasets were generated in these studies. This paper analyzes three existing and publicly available datasets: (1) Data from *Monavarfeshani et al., 2018* can be found at https://elifesciences.org/articles/33498. (2) Data from *Rheaume et al., 2018* can be found at https://health.uconn.edu/neuroregeneration-lab/rgc-subtypes-gene-browser/. (3) Data from *Shekhar et al., 2022* can be found at https://elifesciences.org/articles/73809.

The following previously published datasets were used:

| Author(s) | Year | Dataset title | Dataset URL | Database and Identifier |
|---|---|---|---|---|
| Trakhtenberg E, Rheaume B | 2018 | Single cell transcriptome profiling of retinal ganglion cells identifies cellular subtypes | https://www.ncbi.nlm.nih.gov/geo/query/acc.cgi?acc=GSE115404 | NCBI Gene Expression Omnibus, GSE115404 |

*Continued on next page*

*Continued*

| Author(s) | Year | Dataset title | Dataset URL | Database and Identifier |
|---|---|---|---|---|
| Shekhar K, Whitney I, Butrus S, Peng Y, Sanes JR | 2022 | Diversification of multipotential postmitotic mouse retinal Diversification of multipotential postmitotic mouse retinal ganglion cell precursors into discrete types | https://www.ncbi.nlm.nih.gov/geo/query/acc.cgi?acc=GSE185671 | NCBI Gene Expression Omnibus, GSE185671 |

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
