## [Editor Report]

This study address an interesting mechanistic question with important implications for fundamental neural development. The authors' findings support a model in which retinal ganglion cell axons secrete Shh in the visual thalamus to induce FGF15 expression by astrocytes, which in turn attracts migrating Gad1-expressing cells (interneurons) into the vLGN and dLGN during mouse development. Interestingly, neuronal activity is not required for this process. These findings will be appreciated by a wide range of developmental neuroscientists.

---

## [Decision Letter]

**Decision letter after peer review:**

Thank you for submitting your article "Sonic hedgehog-dependent recruitment of GABAergic interneurons into the developing visual thalamus" for consideration by *eLife*. Your article has been reviewed by 2 peer reviewers, and the evaluation has been overseen by David Ginty as the Reviewing Editor and Marianne Bronner as the Senior Editor. The following individuals involved in review of your submission have agreed to reveal their identity: Andrew M Garrett (Reviewer #1); David Feldheim (Reviewer #2).

Essential revisions:

1) The authors did not show RGC axons targeting to the LGN in SHH mutants (Nes-Cre or Calb2-Cre). Peng et al. (2018) reported that SHH produced by RGCs contributes to axon guidance at the optic chiasm. If axons do not target correctly in these mutants, one cannot rule out the contribution of other axon-derived molecules. Ocular CTb injections in Calb2-Cre SHH F/F mice showing axons in the LGN would go a long way towards alleviating this concern, even if e.g., eye-specific segregation was not entirely normal due to guidance defects at the optic chiasm. The Reviewers and Reviewing Editor agree that this is an important experiment that needs to be done prior to publication.

2) One important consideration is Calb2-Cre expression in the developing thalamus, and the specificity of Calb2-Cre line with respect to RGCs and neurons in the thalamus needs to be experimentally confirmed. Is Calb2-Cre RGC selective or does it remove Shh from regions that contain either stem cells or interneurons before they reach the dLGN and/or thalamic sources of Shh? From the discussion "In Shhfl/fl;Calb2Cre mice, SHH is not lost from these progenitor zones, allowing us to differentiate between progenitor zone-derived SHH functions and RGC-derived SHH functions." This is an important statement and the authors should include this in their analysis of the Calb2-Cre, Tdt reporter in Supp. Figure 1.

One way to ask whether it is loss of Shh from RGCs that results in the phenotype is to rescue the phenotype by expressing Shh in RGCs of the mutant. An AAV injection would enable Shh to be expressed by P8, which should be enough time to rescue. While the reviewers agree that this experiment would not be a requirement for publication, it is a recommendation that the authors may want to consider.

3) The authors provide evidence that they have blocked neural activity, as determined indirectly using eye-specific segregation as readout. However mice still have a diminished but not absent looming response (suggesting that they are not blind). In the absence of any direct measurement of retinal/visual activity, the authors need to be careful in their interpretation of their results. For example in the discussion " Our results revealed that retinal activity had little effect on the percentage of interneurons recruited into dLGN and vLGN" This should be reworded to: Our results revealed that decreasing the level of retinal activity had little effect on the percentage of interneurons recruited into dLGN and vLGN.

4) The case for RGC-derived SHH would be strengthened by demonstrating that it makes it all the way down the axon and into the thalamus. This could be accomplished by immunostaining for SHH in WT vs. Math5 KO LGN. This experiment may not be feasible in the time for resubmission, but this issue should be discussed.

---

## [Author Response]

Essential revisions:1) The authors did not show RGC axons targeting to the LGN in SHH mutants (Nes-Cre or Calb2-Cre). Peng et al. (2018) reported that SHH produced by RGCs contributes to axon guidance at the optic chiasm. If axons do not target correctly in these mutants, one cannot rule out the contribution of other axon-derived molecules. Ocular CTb injections in Calb2-Cre SHH F/F mice showing axons in the LGN would go a long way towards alleviating this concern, even if e.g., eye-specific segregation was not entirely normal due to guidance defects at the optic chiasm. The Reviewers and Reviewing Editor agree that this is an important experiment that needs to be done prior to publication.

This is an excellent point. Indeed, elegant work by Peng et al. (2018) shows that the RGC-derived SHH plays a critical role in axon guidance at the optic chiasm. To determine how RGC axons target the LGN in *Shh^fl/fl^;Calb2^Cre^* mice, we have now performed ocular injection of fluorescently conjugated CTB to analyze contra- and ipsi- RGC projections, as suggested by the reviewers and reviewing editor. We observed that despite a significant decrease in SHH expression in the *Shh^fl/fl^;Calb2^Cre^* mutant retina (Figure 3), RGC axons from both eyes remain capable of innervating the LGN. Quantitative analysis of this experiment revealed no significant difference in the percentage of dLGN area covered by RGC projections (either the contra- or ipsi-RGC projections) between controls and mutants. We think this new data alleviates the concerns raised by the reviewers. This new data has been added as a new Figure 4 and we describe these findings in the revised Results section of the manuscript.

2) One important consideration is Calb2-Cre expression in the developing thalamus, and the specificity of Calb2-Cre line with respect to RGCs and neurons in the thalamus needs to be experimentally confirmed. Is Calb2-Cre RGC selective or does it remove Shh from regions that contain either stem cells or interneurons before they reach the dLGN and/or thalamic sources of Shh? From the discussion "In Shhfl/fl;Calb2Cre mice, SHH is not lost from these progenitor zones, allowing us to differentiate between progenitor zone-derived SHH functions and RGC-derived SHH functions." This is an important statement and the authors should include this in their analysis of the Calb2-Cre, Tdt reporter in Supp. Figure 1.

We also agree with this concern. To address this, we used immunohistochemistry in wild-type mice to study the expression of CALB2 in the developing LGN and progenitor zones during the time of interneuron migration. We did a poor job of describing our previous studies that assessed the developmental expression of CALB2 in visual thalamus (Su et al. 2011), which showed very sparse cellular expression of CALB2 in the neonatal vLGN and a lack of labeled cell bodies in dLGN (although retinal terminals in this region are robustly labeled by CALB2-IHC). We have revised the text to better describe these prior studies, and prior studies where we have used this Calb2-Cre previously (Kerr et al. 2019). We have added the new data as Figure 3—figure supplement 2. We do see some CALB2 cells in the thalamic and tectal progenitor zones, but they are sparse (and based on their distribution do not appear to be interneurons that will eventually migrate into visual thalamus; see Su et al. 2020). As there are some CALB2+ cells in the progenitor zones we have also revised and softened the quoted statement above. That said, we feel that the data regarding the loss of *Fgf15* expression in visual thalamus of *Shh^fl/fl^;Calb2^Cre^* mutants and our prior work using *Fgf15*-null mice support the overall claim of the manuscript that retina-derived SHH is important for driving the expression of this mitogen and recruiting interneurons into the developing visual thalamus. Moreover, we have added new data demonstrating that the loss of SHH in the *Shh^fl/fl^;Calb2^Cre^* mutants does not impact the number or distribution of astrocytes in visual thalamus and it is a subpopulation of these cells that are generating *Fgf15* to attract migrating interneurons (this new data has been added to Figure 5). To us, this data further supports our over-arching hypothesis.

We will point out that a complicated aspect in studying GABAergic neurons in vLGN is that these represent both local interneurons (which are relatively sparse) and principal projection neurons (which are the main cell type in vLGN). Differences in the distribution of *Gad1*-expressing cells in *Shh^fl/fl^;Nes^Cre^* and *Shh^fl/fl^;Calb2^Cre^* mutants suggest to us that the principle cells of vLGN likely do require progenitor zone-derived SHH for their generation, but these are not the cells that we focus on in this study (and the fact that they remain in the *Shh^fl/fl^;Calb2^Cre^* mutants suggest that this line does little to impact progenitor zone derived SHH).

In response to concerns about CALB2 expression, we also did a more thorough job in the revised manuscript of assessing the temporal expression of *Calb2* in embryonic and perinatal retina. Even though the reviewers did not specifically ask for that here, we think it was equally important for the use of this Cre driver line. We performed developmental analysis of *Calb2* expression in the single-cell RNA-seq data from Shekhar et al. (2022). We observed that RGCs begin to significantly upregulate express *Calb2* at E16. This data has been added to Figure 3.

One way to ask whether it is loss of Shh from RGCs that results in the phenotype is to rescue the phenotype by expressing Shh in RGCs of the mutant. An AAV injection would enable Shh to be expressed by P8, which should be enough time to rescue. While the reviewers agree that this experiment would not be a requirement for publication, it is a recommendation that the authors may want to consider.

We agree with the reviewers that AAV injection driving the expression of Shh (or Fgf15) to rescue the thalamic interneuron deficit phenotype will further strengthen our findings. While we agree, we view such experiments as beyond the scope of the current study as such tools are not presently in hand.

3) The authors provide evidence that they have blocked neural activity, as determined indirectly using eye-specific segregation as readout. However mice still have a diminished but not absent looming response (suggesting that they are not blind). In the absence of any direct measurement of retinal/visual activity, the authors need to be careful in their interpretation of their results. For example in the discussion " Our results revealed that retinal activity had little effect on the percentage of interneurons recruited into dLGN and vLGN" This should be reworded to: Our results revealed that decreasing the level of retinal activity had little effect on the percentage of interneurons recruited into dLGN and vLGN.

We agree with the reviewer and therefore, have toned down our claims from these experiments.

4) The case for RGC-derived SHH would be strengthened by demonstrating that it makes it all the way down the axon and into the thalamus. This could be accomplished by immunostaining for SHH in WT vs. Math5 KO LGN. This experiment may not be feasible in the time for resubmission, but this issue should be discussed.

We agree with the reviewers that this is an important experiment to do, and in fact we previously purchased antibodies to test this. Unfortunately, those antibodies failed to detect endogenous SHH in our hands. After receiving these reviews, we used social media to reach out widely to colleagues studying SHH (beyond our own network of collaborators) to determine if there were better (or any) commercial antibodies that detect SHH. We universally heard that no such antibody existed and in fact we had used the best antibody available. Since this antibody is a monoclonal we thought perhaps we had purchased a bad batch. Therefore, we contacted the Developmental Studies Hybridoma Bank and ordered a new aliquot (and different batch) of the 5E1 anti-SHH antibody. Unfortunately, it was equally ineffective in labeling SHH in WT mice.

Nevertheless, other studies have reported that RGC-derived SHH protein is present in the rodent optic nerve (Wallace and Raff 1999) and travels as far as the superior colliculus (Traiffort et al. 2001). An alternative approach for establishing active SHH signaling that several labs use as a proxy for detecting SHH protein, is to test for the presence of Gli1. In fact, one of our co-authors previously reported that loss of Shh abrogates Gli1 expression, indicating that it is a reliable readout of Shh activity (Garcia et al. 2010). Here, by staining for βGal in P3 Gli1lacZ mice, we demonstrate that there is indeed active SHH signaling in LGN during the time of interneuron migration. We have now added this data as Figure 3—figure supplement 1.